# Effect of growing regions on morphological characteristics, protein subfractions, rumen degradation and molecular structures of various whole-plant silage corn cultivars

Xinyue Zhang[1,2☯], Nazir Ahmad Khan[3☯], Enyue Yao[1¤a], Fanlin Kong[2], Ming Chen[4], Rifat Ullah Khan[3], Xin Liu[1¤b], Yonggen Zhang[1], Hangshu Xin[1] *

1 College of Animal Science and Technology, Northeast Agricultural University, Harbin, China, 2 State Key Laboratory of Animal Nutrition, Beijing Engineering Technology Research Center of Raw Milk Quality and Safety Control, College of Animal Science and Technology, China Agricultural University, Beijing, China, 3 Department of Animal Nutrition, The University of Agriculture Peshawar, Peshawar, Khyber Pakhtunkhwa, Pakistan, 4 College of Agronomy and Biotechnology, National Maize Improvement Center of China, China Agricultural University, Beijing, China

☯ These authors contributed equally to this work.
¤a Current address: Gushi Biological Group, Harbin, China
¤b Current address: Wellhope Shenyang Ruminant Feed Company Limited, Shenyang, China
* hangshuxin@neau.edu.cn

**Data Availability Statement:** All relevant data for this study are available in the figshare repository

## Abstract

Little information exists on the variation in morphological characteristics, nutritional value, ruminal degradability, and molecular structural makeup of diverse whole-plant silage corn (WPSC) cultivars among different growing regions. This study investigated the between-regions (Beijing, Urumchi, Cangzhou, Liaoyuan, Tianjin) discrepancies in five widely used WPSC cultivars in China (FKBN, YQ889, YQ23, DK301 and ZD958), in terms of 1) morphological characteristics; 2) crude protein (CP) chemical profile; 3) Cornell Net Carbohydrate and Protein System (CNCPS) CP subfractions; 4) in situ CP degradation kinetics; and 5) CP molecular structures. Our results revealed significant growing region and WPSC cultivar interaction for all estimated morphological characteristics ($P < 0.001$), CP chemical profile ($P < 0.001$), CNCPS subfractions ($P < 0.001$) and CP molecular structural features ($P < 0.05$). Except ear weight ($P = 0.18$), all measured morphological characteristics varied among different growing regions ($P < 0.001$). Besides, WPSC cultivars planted in different areas had remarkably different CP chemical profiles and CNCPS subfractions ($P < 0.001$). All spectral parameters of protein primary structure of WPSC differed ($P < 0.05$) due to the growing regions, except amide II area ($P = 0.28$). Finally, the area ratio of amide I to II was negatively correlated with the contents of soluble CP ($\delta = -0.66$; $P = 0.002$), CP ($\delta = -0.61$; $P = 0.006$), non-protein nitrogen ($\delta = -0.56$; $P = 0.004$) and acid detergent insoluble CP ($\delta = -0.43$; $P = 0.008$), in conjunction with a positive correlation with moderately degradable CP ($PB_1$; $\delta = 0.58$; $P = 0.01$). In conclusion, the cultivar of DK301 exhibited high and stable CP content. The WPSC planted in Beijing showed high CP, SCP and NPN. The low rumen degradable protein of WPSC was observed in

([https://doi.org/10.6084/m9.figshare.22547926.v1](https://doi.org/10.6084/m9.figshare.22547926.v1)).

**Funding:** This research was funded by the National Natural Science Foundation of China (Grant No. 31702135) and China Agriculture Research System of MOF and MARA. The funders had no role in study design, data collection and analysis, decision to publish, or preparation of the manuscript.

**Competing interests:** The authors have declared that no competing interests exist.

Urumchi. Meanwhile, above changes in protein profiles and digestibility were strongly connected with the ratio of amide I and amide II.

## Introduction

Globally, whole plant silage corn (WPSC) is a major ingredient in dairy total mixed ration (TMR) under most dietary regimes, owing to its high and stable biomass yield under a wide range of environment and soil conditions, in conjunction with its high contents of total digestible nutrients and metabolizable energy [1–3]. Moreover, the inclusion of corn silage in TMR increases dry matter (DM) and metabolizable energy intakes, as well as milk yield of dairy cows [4]. Because of these great qualities, WPSC can contribute up to 42% of the TMR for high-producing dairy cows [5]. Thus, optimizing the biomass yield and nutritional qualities of WPSC is a long-term goal for dairy industry to meet the nutritional requirements of high producing dairy cows on sustainable basis.

With the exception of genotype and harvest maturity, the yield and nutritional quality of WPSC are highly influenced by environmental conditions [6–8]. For instance, high growing temperature reduced the digestibility of corn silage, which was a result of an increase in lignin content of stovers and a decrease in starch content of cobs [9, 10]. A close association was observed between soil moisture and plants' nutrients distribution in eight cultivars of napiergrass [10]. Moreover, Temime et al. [9] reported that besides genetic factors, the proportion of volatile constituents in Chétouiolive oils was strongly influenced by growing regions. Similarly, the glycosidase inhibitory rate in mulberry leaves ranged from 142 to 845 g/kg between different cultivated regions [11].

Although, a lower content of crude protein (CP), 74 g/kg DM on average was reported for corn silage, it significantly contributed to the overall supply of metabolizable protein to dairy cows because of high consumption [2]. Moreover, rumen protein degradability and overall supply of metabolizable protein to dairy cows were strongly influenced by variations of CP chemical profiles and protein inherent molecular structures in feedstuffs [4, 12]. Recent studies suggested that the molecular structure of roughages could be revealed by advanced vibrational molecular spectroscopy techniques, for example, attenuated total reflectance-Fourier transform infrared spectroscopy (ATR-FTIR) [13–15]. Thus, it is possible to associate the protein chemical profiles and *in situ* degradation parameters with the spectral data of specific inherent structures [16–19].

To our knowledge, no systematic study is conducted on morphological characteristics, CP rumen degradability and molecular structures of various WPSC cultivars planted in different growing regions. Therefore, in the present study, five widely utilized WPSC cultivars (FKBN, YQ889, YQ23, DK301 and ZD958) planted in five growing regions (Beijing, Urumchi, Cangzhou, Tianjin and Liaoyuan) in China were evaluated on the basis of 1) morphological traits; 2) CP chemical profile; 3) Cornell Net Carbohydrate and Protein System (CNCPS) CP subfractions; 4) *in situ* CP degradation kinetics; and 5) protein molecular structures. The relationships between protein molecular structural parameters and CP chemical profile, CNCPS subfractions as well as ruminal degradation characteristics were also investigated. We hypothesized that the morphological characteristics, CP chemical profile, and ruminal degradability may differ in WPSC cultivated in different growing regions, and those discrepancies in protein nutrition and rumen degradability can be correlated with their inherent protein molecular structures.

## Materials and methods

### Corn cultivars, growing regions, and crop production

Five WPCS cultivars widely utilized for silage production in China were selected for this study, including Fengkenbainuo (FKBN), Yaqing 889 (YQ889), Yuqing 23 (YQ23), Dongke 301 (DK301) and Zhengdan 958 (ZD958). All cultivars were grown in five different regions, which were Beijing (39˚9' N, 116˚3' E), Urumchi (43˚8' N, 87˚7' E), Cangzhou (38˚3' N, 116˚8' E), Tianjin (39˚1' N, 117˚2' E) and Liaoyuan (42˚7' N, 125˚5' E). In each region, every cultivar was sown in three plots (5 m × 5 m) with seeding rate of 1400 kernels per plot, with plant-to-plant distance of 35 cm and row to row distance of 60 cm. The fertilization and irrigation were managed by the local farmers accordingly. Data on geographical and climate condition of the five experimental regions are summarized in Table 1.

### Crop sampling and morphological measurements

Twenty plants (10 cm above the ground) from each plot were randomly harvested at kernel maturity stage of half milk line from September to October in 2017. The exterior 1 m area of each plot was excluded from sampling to ensure uniformity in the sampled plants. The average value of nutrient indicators from twenty WPCS plants was seemed as raw data, three plots were treated as replicates. The physical parameters of WPCS including whole-plant height, whole-plant weight, stem diameter, ear height, ear weight, ear length, ear diameter, ear rows and kernel numbers in each row were measured and recorded at harvest. Then, the whole-plant samples were chopped to small particle size (1–2 cm), dried (at 60 ˚C for 24 h) and grounded through a 1.0 mm screen (FZ102, Taisite Instrument Co., Ltd, Tianjin, China) for analysis of chemical components and CNCPS compositions, or through a 2 mm screen for *in situ* incubation measurements, or through a 0.25 mm screen for molecular spectral analysis of protein primary and secondary structures.

**Table 1. Longitude and latitude, average annual temperature, and average annual precipitation of different growing regions of whole-plant silage corn cultivars.**

| Region | Longitude | Latitude | Temperature (˚C) | Precipitation (mm) |
|---|---|---|---|---|
| *Beijing* | E 116˚3' | N 39˚9' | July: 23~31 | July: 97.5 |
| | | | August: 21~30 | August: 233.9 |
| | | | September: 17~27 | September: 2.8 |
| | | | October: 7~15 | October: 73.3 |
| *Urumchi* | E 87˚7' | N 43˚8' | July: 21~31 | July: 3.2 |
| | | | August: 17~26 | August: 14.5 |
| | | | September: 11~22 | September: 3.7 |
| | | | October: 2~11 | October: 8.8 |
| *Cangzhou* | E 116˚8' | N 38˚3' | July: 24~32 | July: 161.0 |
| | | | August: 22~30 | August: 124.7 |
| | | | September: 18~29 | September: 4.7 |
| | | | October: 8~16 | October: 138.6 |
| *Liaoyuan* | E 125˚5' | N 42˚7' | July: 19~30 | July: 231.8 |
| | | | August: 17~27 | August: 259.0 |
| | | | September: 9~23 | September: 16.7 |
| | | | October: 0~12 | October: 1.6 |
| *Tianjin* | E 117˚2' | N 39˚1' | July: 25~32 | July: 159.6 |
| | | | August: 23~31 | August: 160.4 |
| | | | September: 19~29 | September: 1.2 |
| | | | October: 9~17 | October: 101.9 |

## Protein chemical analysis and subfractions composition

The WPSC samples were analyzed for CP content (official method 984.13, using a Kjeltec 2400 autoanalyzer; Foss Analytical A/S, Hillerϕd, Denmark) according to AOAC [20]. The contents of soluble CP (SCP), neutral detergent insoluble CP (NDICP) and acid detergent insoluble CP (ADICP) were determined according to Licitra et al. [21].

For estimating the difference of CP degradation in the rumen, the CP subfractions composition were determined according to the updated version (v 6.5) of Cornell Net Carbohydrate Protein System [CNCPS, 22, 23]. A total of five subfractions, quantified in this study, were: $PA_1$ (non-protein N; ruminal degradation rate ($k_d$) of 200%/h), $PA_2$ (soluble non-ammonia CP; $k_d$ of 10%-40%/h), $PB_1$ (moderately degradable CP; $k_d$ of 3%-20%/h), $PB_2$ (slowly degradable CP; $k_d$ of 1%-18%/h) and PC (unavailable CP).

These CP subfractions were estimated from the chemical profiles according to the following mathematical models:

$$PA_1(g/kg\,CP) = NPN(g/kg\,SCP) \times 0.01 \times SCP(g/kg\,CP)$$

$$PA_2(g/kg\,CP) = SCP(g/kg\,SCP) - PA_1(g/kg\,CP)$$

$$PB_1(g/kg\,CP) = 1000 - PA_1(g/kg\,CP) - PA_2(g/kg\,CP) - PB_2(g/kg\,CP) - PC(g/kg\,CP)$$

$$PB_2(g/kg\,CP) = NDICP(g/kg\,CP) - ADICP(g/kg\,CP)$$

$$PC\,(g/kg\,CP) = ADICP(g/kg\,CP)$$

## In situ rumen degradation

Three healthy and non-lactating Holstein Frisian dairy cows (550–600 kg body weight), fixed with permanent rumen fistula in the experimental base of Northeast Agricultural University (Approval number: NEAU-[2011]-9) were used for *in situ* incubation research. These animals were fed TMR with forage to concentrate ratio of 60:40, twice daily at 07:00 and 21:00 h. The ingredients and nutrient contents of TMR are list in Table 2 according to recommendations in NRC (2001) guidelines [24].

Subsamples (ca. 7 g) were randomly incubated in sealed nylon bags (10 × 20 cm, pore size 40 μm) in rumen of three fistulated cows for 0, 4, 8, 12, 24, 36, 48 and 72 h, using the "gradual in/all out" schedule. Three replicate bags per sample were used for every incubation time point. After incubation, all nylon bags were taken out of the rumen, washed with cold running tap water for six times, and then dried to constant weight in forced air over at 65 ˚C. The dried residues from replicate bags of each sample were pooled according to the incubation time, then grounded and stored in plastic sealed bags for further analysis. The whole procedure was conducted twice. The parameters related to rumen degradation characteristics of CP were calculated according to Ørskov and McDonald [25].

## Spectral data collection and analysis

Data on protein primary and secondary structures were collected at the Chemical Molecular Structure Analysis Laboratory located in Northeast Agriculture University (Harbin, China). Fourier transform infrared spectroscopy (FT/IR; Bruker ALPHA-T, Germany) was used for the molecular spectral analysis. The spectral data were obtained at the mid-infrared fingerprint

**Table 2. Composition and nutrient levels of the basal diet.** (DM basis).

| Items | Content % |
|---|---|
| Ingredients | |
| Chinese wildrye | 42.85 |
| Corn silage | 15.82 |
| Corn | 13.18 |
| Wheat bran | 3.74 |
| Molasses beet | 0.99 |
| Soybean meal | 3.15 |
| Dried distillers grain | 5.35 |
| Cottonseed meal | 2.06 |
| Corn gluten feed | 7.42 |
| Corn germ meal | 4.94 |
| Premix[1] | 0.50 |
| Total | 100.00 |
| Nutrient levels[2] | |
| $NE_L$ / (MJ / kg) | 5.44 |
| CP | 14.40 |
| NDF | 49.20 |
| ADF | 30.60 |
| Ca | 0.60 |
| P | 0.40 |

[1] Contained the following per kg of the premix: vitamin A 8 000 000 IU, Vitamin D 700 000 IU, Vitamin E 10000 IU, Fe 1600 mg, Cu 1500 mg, Zn 10000 mg, Mn 3500 mg, Se 80 mg, I 120 mg, Co 50 mg.

[2], $NE_L$ was a calculated value, while the others were measured values.

region from ca. 4000 to 400 cm$^{-1}$, with 128 coadded scans at a spectral resolution of 4 cm$^{-1}$. Each sample was scanned five times. Data were collected for peak height and area of the spectral bands related to protein primary structural functional groups. Only the spectral intensities of amide I band (baseline: ca. 1712–1565 cm$^{-1}$; peak height: ca. 1639 cm$^{-1}$) and amide II band (baseline: ca. 1565–1489 cm$^{-1}$; peak height: ca. 1545 cm$^{-1}$) were analyzed. The fourier self-deconvolution (FSD) method and the second derivative function in OMNIC 8.2 software (Spectra Tech., Madison, WI, USA) were used to characterize the protein secondary structure in amide I region including protein $\alpha$-helix (peak height: ca. 1656 cm$^{-1}$) and protein $\beta$-sheet (peak height: ca. 1634 cm$^{-1}$). The protein molecular structure spectral data quantified in this study were the peak height and area of protein amide I, amide II and their ratio, in conjunction with peak height of $\alpha$-helix, $\beta$-sheet and their ratio.

To visualize the structural differences in protein structures, multivariate analysis, agglomerative hierarchical cluster analysis (CLA) and principal component analysis (PCA) were applied on the overall spectroscopic data within the protein amide I and amide II region (ca. 1712–1489 cm$^{-1}$), using Statistica 8.0 software (StatSoft Inc., Tulsa, OK, USA). The detailed procedure and concepts of performing CLA and PCA on FTIR spectral data was described by Yu [26, 27].

## Statistical analysis

The PROC MIXED procedure of SAS 9.4 was used to analyze data on the WPSC morphological characteristics, CP chemical profiles, CNCPS subfractions, in situ CP degradation kinetics and protein molecular structural parameters, the model used was: $Y_{ijkl} = \mu + C_i + R_j + C_i \times R_j$

+ $D_k$ +$e_{ijkl}$, where $Y_{ijkl}$ was an observation on the dependent variable ijkl, μ was the population mean of the variable, $C_i$ was a fixed effect of corn cultivar (i = 5; FKBN, YQ889, YQ23, DK301 and ZD958), $R_j$ was a fixed effect of growing region (j = 5; Beijing, Urumchi, Cangzhou, Tianjin and Liaoyuan), $C_i \times R_j$ was a fixed effect of interaction between factor C at level i and factor R at level j, $D_k$ was the random effect, and eijkl was the model error.

The correlation between protein spectral parameters and protein chemical profile, CNCPS subfractions as well *in situ* CP degradation data were analyzed by the Hmisc and Corrplot packages of R (version 4.0.2; R Foundation for Statistical Computing, Vienna, Austria). For all statistical analysis, significance was declared at $P < 0.05$, and tendency was declared at $0.05 < P < 0.10$.

## Results and discussion

### Effects of growing regions and cultivars on morphological characteristics of whole plant silage corn

The significant interaction between cultivar and growing region on morphological characteristics is showed in Table 3 ($P < 0.05$). Except ear weight ($P = 0.18$), all measured morphological characteristics varied ($P < 0.05$) among the growing regions. Plant height and weight, in conjunction with ear height and length were highest ($P < 0.05$) in WPSC cultivars grown in Urumchi, while whole plant height and weight, as well ear length were lowest ($P < 0.05$) in Tianjin. Cultivars grown in Beijing represented greatest ($P < 0.05$) stem diameter, ear diameter and rows of kernel, whereas those in Urumchi showed the lowest values for these parameters ($P < 0.05$). The variation in growth and morphological characteristics of WPSC affects biomass yield, fermentation quality, and nutritional value of silages [28]. Maize plant with greater height and cobs weight are associated with high biomass yield and starch yields, and reflects crop growth attained during the growing period [28]. The effect of growing regions on yield and morphological characteristics of WPSC could be related to differences in soil fertility, growing temperature, and day length, and to the intrinsic variation in the adaptability of the different cultivars to these conditions [2, 28]. Mean number of kernels in a row was highest ($P < 0.05$) in Tianjin and lowest ($P < 0.05$) in Beijing. Previous studies have reported that precipitation was one of the most influential abiotic factors for plant productivity [29], and drought stress generally contributed to delay in plant growth and development by decreasing cell elongation and reducing photosynthesis [30]. Moreover, soil moisture and growing temperature were highly related to DM yields because they affected canopy and anatomical development of corn [10, 31]. Wang et al. [32] reported that the WPSC cultivar of Yunuo_7 had better leaf appearance rate at high growing temperature (37.5 ˚C, southwest of China) than other corn cultivars adopted to northeast and north plain of China. Furthermore, there was an effect of solar radiation on growth components which were proportional to gross photosynthesis [33]. Results of Tollenaar et al. [34] showed that more than a quarter of DM yield of corn was attributable to solar brightening during 1984 to 2013 in the US Corn Belt. In addition, limited solar radiation during flowering reduced kernel number and kernel development of corn [35]. Above factors may partially elucidate the reasons for discrepancy in morphological characteristics of WPSC cultivars from different growing regions observed in our study.

### Effects of growing regions and cultivars on protein chemical profiles and CNCPS of whole plant silage corn

In the present study, WPSC cultivars grown in different areas exhibited remarkable discrepancy on all measured CP chemical profiles and CNCPS subfractions ($P < 0.05$; Table 4).

**Table 3. Morphological measurements of silage-corn cultivars grown in different regions.** %.

| Cultivars | Regions | Whole plant | | Stem | Ear | | | | | Row kernels |
|---|---|---|---|---|---|---|---|---|---|---|
| | | Height (cm) | Weight (kg) | Diameter (cm) | Height (cm) | Weight (kg) | Length (cm) | Diameter (cm) | Rows | |
| FKBN | Beijing | 340.0 | 1.35 | 2.31 | 156.2 | 0.53 | 31.2 | 8.26 | 17.2 | 28.6 |
| | Urumchi | 320.1 | 2.26 | 1.66 | 168.7 | 0.84 | 29.7 | 5.72 | 13.8 | 40.8 |
| | Cangzhou | 317.8 | 0.89 | 1.80 | 151.3 | 0.33 | 27.5 | 5.59 | 13.2 | 28.8 |
| | Tianjin | 297.7 | 0.86 | 1.56 | 151.3 | 0.39 | 24.3 | 5.37 | 14.8 | 39.2 |
| | Liaoyuan | 293.2 | 1.01 | 2.00 | 149.8 | 0.43 | 26.6 | 6.28 | 16.0 | 42.2 |
| YQ889 | Beijing | 316.0 | 0.96 | 2.09 | 152.5 | 0.36 | 24.4 | 5.92 | 16.6 | 30.1 |
| | Urumchi | 363.6 | 1.17 | 1.84 | 190.8 | 0.35 | 28.0 | 5.89 | 15.2 | 30.6 |
| | Cangzhou | 328.0 | 1.31 | 1.93 | 155.1 | 0.50 | 27.1 | 6.16 | 15.4 | 30.7 |
| | Tianjin | 254.2 | 1.99 | 1.75 | 122.0 | 0.72 | 23.5 | 5.96 | 15.6 | 33.6 |
| | Liaoyuan | 269.5 | 0.77 | 1.90 | 127.3 | 0.33 | 24.9 | 5.84 | 15.6 | 31.8 |
| YQ23 | Beijing | 280.4 | 1.78 | 2.20 | 124.7 | 0.83 | 24.8 | 6.33 | 16.4 | 34.9 |
| | Urumchi | 345.5 | 0.96 | 1.59 | 186.3 | 0.31 | 29.3 | 5.64 | 13.8 | 24.2 |
| | Cangzhou | 330.4 | 1.15 | 1.68 | 154.8 | 0.44 | 25.1 | 5.86 | 16.2 | 32.8 |
| | Tianjin | 262.9 | 0.66 | 1.79 | 143.2 | 0.26 | 23.3 | 5.68 | 14.8 | 32.1 |
| | Liaoyuan | 296.0 | 1.21 | 2.12 | 154.0 | 0.44 | 26.1 | 6.04 | 14.4 | 37.8 |
| DK301 | Beijing | 301.6 | 0.87 | 2.01 | 142.4 | 0.33 | 23.2 | 6.28 | 17.2 | 27.4 |
| | Urumchi | 351.0 | 1.32 | 1.97 | 182.8 | 0.42 | 31.6 | 6.34 | 14.0 | 33.7 |
| | Cangzhou | 282.9 | 1.93 | 1.71 | 131.8 | 0.80 | 23.4 | 6.24 | 14.6 | 32.2 |
| | Tianjin | 257.6 | 0.88 | 1.98 | 124.0 | 0.41 | 24.7 | 6.42 | 16.0 | 39.8 |
| | Liaoyuan | 298.6 | 1.02 | 1.69 | 158.1 | 0.33 | 27.3 | 5.72 | 14.8 | 34.8 |
| ZD958 | Beijing | 271.8 | 0.80 | 1.87 | 137.6 | 0.33 | 24.0 | 6.15 | 15.8 | 26.7 |
| | Urumchi | 383.3 | 1.19 | 1.62 | 208.9 | 0.34 | 28.1 | 4.85 | 13.8 | 36.0 |
| | Cangzhou | 301.4 | 1.28 | 2.00 | 137.4 | 0.46 | 26.2 | 6.47 | 14.0 | 36.5 |
| | Tianjin | 340.5 | 1.08 | 2.32 | 185.3 | 0.53 | 23.9 | 6.41 | 17.7 | 40.3 |
| | Liaoyuan | 260.7 | 1.76 | 1.84 | 117.8 | 0.72 | 24.3 | 6.00 | 14.8 | 37.1 |
| | SEM | 69.04 | 0.82 | 0.42 | 46.54 | 0.34 | 4.85 | 1.16 | 2.36 | 9.36 |
| Regions | | | | | | | | | | |
| | Beijing | 301.96[c] | 1.15[b] | 2.10[a] | 142.68[b] | 0.48[ab] | 25.52[b] | 6.59[a] | 16.64[a] | 29.54[c] |
| | Urumchi | 352.70[a] | 1.38[a] | 1.74[c] | 187.50[a] | 0.45[b] | 29.34[a] | 5.69[c] | 14.12[d] | 33.06[b] |
| | Cangzhou | 312.10[b] | 1.31[a] | 1.82[bc] | 146.08[b] | 0.51[a] | 25.86[b] | 6.06[b] | 14.68[cd] | 32.20[b] |
| | Tianjin | 282.58[d] | 1.09[b] | 1.88[b] | 145.16[b] | 0.46[ab] | 23.94[c] | 5.97[b] | 15.78[b] | 37.00[a] |
| | Liaoyuan | 283.60[d] | 1.15[b] | 1.91[bc] | 141.40[b] | 0.45[b] | 25.84[b] | 5.98[b] | 15.12[bc] | 36.74[a] |
| | SEM | 24.71 | 0.44 | 0.38 | 18.75 | 0.19 | 3.60 | 0.71 | 2.45 | 8.81 |
| P-value | Regions | <0.001 | <0.001 | <0.001 | <0.001 | 0.18 | <0.001 | <0.001 | <0.001 | <0.001 |
| | Cultivar | <0.001 | 0.38 | 0.75 | 0.002 | 0.29 | <0.001 | 0.001 | 0.35 | 0.001 |
| | Region × Cultivar | <0.001 | <0.001 | <0.001 | <0.001 | <0.001 | <0.001 | <0.001 | <0.001 | <0.001 |

Note: FKBN = Fengkenbainuo; YQ 889 = Yaqing 889; YQ 23 = Yuqing 23; DK 301 = Dongke 301; ZD958 = Zhengdan 958. Means with different letters are significantly different (P < 0.05).

The CP content (g/100 g) was greatest in Beijing (8.66), followed by Tianjin (8.57), Cangzhou (7.82), Liaoyuan (7.80) and Urumchi (7.35). The CP content of the maize silages in the current study was higher than the range of CP values (6.3 to 7.5 g/100 g) reported for different maize genotypes [28], which could be related to the early harvest stage selected in the present

**Table 4. Crude protein (CP) chemical components and the CNCPS subfractions of whole plant silage corn cultivars grown in different regions.**

| Cultivars | Regions | CP components (g/100 g DM) | | | | | CNCPS subfractions (g/100 g CP) | | | | |
|---|---|---|---|---|---|---|---|---|---|---|---|
| | | CP | NPN | SCP | NDICP | ADICP | $PA_1$ | $PA_2$ | $PB_1$ | $PB_2$ | PC |
| FKBN | Beijing | 8.75 | 1.85 | 2.22 | 1.80 | 0.70 | 21.18 | 4.20 | 54.01 | 12.62 | 7.99 |
| | Urumchi | 6.95 | 1.02 | 1.66 | 1.65 | 0.37 | 14.66 | 9.27 | 52.36 | 18.33 | 5.39 |
| | Cangzhou | 8.21 | 1.26 | 1.89 | 2.01 | 0.61 | 15.29 | 7.66 | 52.57 | 17.03 | 7.46 |
| | Tianjin | 8.91 | 2.06 | 2.72 | 1.70 | 0.38 | 23.06 | 7.40 | 50.52 | 14.78 | 4.24 |
| | Liaoyuan | 7.07 | 0.89 | 1.39 | 1.46 | 0.35 | 12.57 | 7.05 | 59.74 | 15.68 | 4.96 |
| YQ889 | Beijing | 8.48 | 1.87 | 2.38 | 2.13 | 0.57 | 22.07 | 6.01 | 46.86 | 18.37 | 6.69 |
| | Urumchi | 7.74 | 1.94 | 2.57 | 1.86 | 0.53 | 25.09 | 8.07 | 42.87 | 17.06 | 6.91 |
| | Cangzhou | 6.95 | 1.55 | 1.92 | 2.34 | 0.57 | 22.11 | 5.35 | 38.89 | 25.50 | 8.16 |
| | Tianjin | 8.01 | 0.86 | 1.63 | 2.83 | 0.59 | 10.69 | 9.64 | 44.37 | 27.93 | 7.37 |
| | Liaoyuan | 8.12 | 1.03 | 2.04 | 1.79 | 0.55 | 12.63 | 12.46 | 52.86 | 15.24 | 6.82 |
| YQ23 | Beijing | 9.08 | 1.72 | 2.25 | 1.90 | 0.40 | 18.95 | 5.85 | 54.27 | 16.55 | 4.38 |
| | Urumchi | 6.90 | 1.24 | 1.91 | 1.39 | 0.37 | 17.93 | 9.70 | 52.19 | 14.81 | 5.37 |
| | Cangzhou | 8.48 | 1.36 | 1.76 | 2.92 | 0.59 | 15.97 | 4.72 | 44.82 | 27.59 | 6.90 |
| | Tianjin | 9.09 | 1.76 | 2.51 | 2.25 | 0.52 | 19.34 | 8.22 | 47.68 | 19.07 | 5.69 |
| | Liaoyuan | 8.20 | 1.10 | 1.72 | 2.19 | 0.59 | 13.38 | 7.60 | 52.27 | 19.62 | 7.14 |
| DK301 | Beijing | 9.14 | 1.57 | 2.25 | 2.17 | 0.42 | 17.16 | 7.37 | 51.77 | 19.15 | 4.55 |
| | Urumchi | 8.26 | 1.50 | 1.42 | 1.55 | 0.40 | 18.15 | 5.46 | 57.62 | 13.99 | 4.78 |
| | Cangzhou | 8.79 | 1.16 | 2.00 | 2.31 | 0.39 | 13.12 | 9.51 | 51.12 | 21.84 | 4.39 |
| | Tianjin | 8.54 | 0.73 | 1.33 | 2.38 | 0.47 | 8.56 | 6.98 | 56.61 | 22.38 | 5.48 |
| | Liaoyuan | 8.06 | 1.06 | 1.72 | 1.76 | 0.45 | 13.12 | 8.18 | 56.94 | 16.21 | 5.56 |
| ZD958 | Beijing | 7.85 | 1.59 | 1.91 | 1.48 | 0.31 | 20.26 | 4.05 | 56.88 | 14.91 | 3.90 |
| | Urumchi | 6.88 | 1.26 | 1.63 | 1.19 | 0.15 | 18.32 | 5.41 | 58.99 | 15.11 | 2.18 |
| | Cangzhou | 6.66 | 0.93 | 1.44 | 1.24 | 0.39 | 14.00 | 7.58 | 59.78 | 12.82 | 5.82 |
| | Tianjin | 8.28 | 1.39 | 1.97 | 1.45 | 0.35 | 16.71 | 6.98 | 58.90 | 13.15 | 4.26 |
| | Liaoyuan | 7.54 | 12.3 | 1.76 | 1.48 | 0.31 | 16.30 | 7.02 | 57.14 | 15.48 | 4.05 |
| SEM | | 0.151 | 0.073 | 0.074 | 0.090 | 0.025 | 0.807 | 0.381 | 1.104 | 0.844 | 0.292 |
| Regions | | | | | | | | | | | |
| | Beijing | 86.6[a] | 17.2[a] | 22.0[a] | 19.0[b] | 4.8[ab] | 199.2[a] | 55.0[c] | 527.6[b] | 163.2[c] | 55.0[b] |
| | Urumchi | 73.5[c] | 13.9[b] | 18.4[b] | 15.3[d] | 3.6[c] | 188.3[a] | 75.8[ab] | 528.1[b] | 158.6[c] | 49.3[c] |
| | Cangzhou | 78.2[b] | 12.5[c] | 18.0[c] | 21.6[a] | 5.1[a] | 161.0[b] | 69.6[b] | 494.4[c] | 209.6[a] | 65.5[a] |
| | Tianjin | 85.7[a] | 13.6[bc] | 20.3[b] | 21.2[a] | 4.6[b] | 156.7[b] | 78.4[ab] | 516.2[b] | 194.6[b] | 54.1[b] |
| | Liaoyuan | 78.0[b] | 10.6[d] | 17.3[c] | 17.4[c] | 4.5[b] | 136.0[c] | 84.6[a] | 557.9[a] | 164.5[c] | 57.1[b] |
| SEM | | 0.55 | 0.43 | 0.41 | 0.40 | 0.012 | 4.56 | 3.43 | 6.37 | 3.58 | 1.30 |
| P-value | Region | < .001 | < .001 | < .001 | < .001 | < .001 | < .001 | < .001 | < .001 | < .001 | < .001 |
| | Cultivar | < .001 | < .001 | < .001 | < .001 | < .001 | < .001 | 0.002 | < .001 | < .001 | < .001 |
| | Region × Cultivar | < .001 | < .001 | < .001 | < .001 | < .001 | < .001 | < .001 | < .001 | < .001 | < .001 |

Note: FKBN = Fengkenbainuo; YQ 889 = Yaqing 889; YQ 23 = Yuqing 23; DK 301 = Dongke 301; ZD958 = Zhengdan 958; 2 CP = crude protein; NPN = non-protein nitrogen; SCP = soluble crude protein; NDICP = neutral detergent insoluble crude protein; ADICP = acid detergent insoluble crude protein; 3 $PA_1$ = ammonia; $PA_2$ = soluble nonammonia CP; $PB_1$ = moderately degradable CP; $PB_2$ = slowly degradable CP; PC = unavailable CP. Means with different letters are significantly different ($P < 0.05$).

study. Nevertheless, the CP values were within the range (5.7 to 12.4 g/100 g CP) reported for WPSC in the meta-analysis of Khan et al. 2015 [2] Meanwhile, the soluble protein contents, such as NPN and SCP, of WPSC planted in Beijing were higher ($P < 0.05$) than other regions. The results indicated that the WPSC grown in Beijing had a high proportion of protein

available to animals, and implied that climate conditions (e.g. precipitation and growing temperature) could influence internal nutrient accumulations of WPSC. In agreement with our findings, previous study reported that the CP contents in corn and wheat were lower in rainy years [36]. Besides, the highest CP content was recorded for DK301 (85.6 g/kg) and lowest for ZD958 (7.44 g/kg). In addition, the greatest variation range of CP among different regions was found for YQ23 cultivar (6.90 to 9.09 g/100 g) and the lowest for DK301 (8.06 to 9.14 g/100 g). These results demonstrated that the cultivar of DK301 had great and stable CP quality, which might be explained by genetic composure [37, 38]. Similar variation in CP content among maize genotypes have been reported earlier studies [8, 28].

For ruminants, the nutritive value of CP in feeds not only depends on the amino acid content and composition but also on the CP degradability (rate and extent) in the rumen, microbial protein synthesis from the degraded protein, and intestinal digestibility of rumen undegraded feed protein, which greatly affect the supply of metabolizable protein to the animal [6]. The CNCPS is widely used for the evaluation of CP in feedstuff for ruminants, based on CP subfractions, characterized on differences in chemical composition and digestibility [39–42]. In our research, the highest $PA_1$ concentration (19.92 g/kg) and lowest PC (4.93 g/kg) were observed ($P < 0.05$) in WPSC cultivated in Urumchi. The results suggested that WPSC grown in Urumchi contained less unavailable protein content (associated with lignin and tannins) than the others [43]. High growth temperature markedly increased the lignin content of corn plants [11, 44], which could explain the lowest PC content in WPSC grown in Urumchi area because of its lowest temperature.

## Effects of growing regions and cultivars on protein in situ rumen degradation of whole plant silage corn

A significant interaction of corn cultivar and growing region was observed in *in situ* soluble CP (S; $P < 0.05$) and ruminal degradable CP (RDP; $P < 0.05$). However, the interaction was not significant ($P > 0.05$) in the rate of degradation ($K_d$), potentially ruminal degradable CP (D), undegradable CP (U), and ruminal undegradable CP (RUP) (Table 4). Among *in situ* CP degradability characteristics, the contents of $K_d$ and RUP varied ($P < 0.05$) among growing regions (Table 5). The highest ($P < 0.05$) content of RUP was observed in the WPSC grown in Beijing (41.7 g/kg DM), followed by Tianjin (39.3 g/kg DM), Liaoyuan (34.9 g/kg DM), Urumchi (34.7 g/kg DM) and Cangzhou (33.0 g/kg DM). The variation of RDP in WPSC from different regions was highest in FKBN (39.3 to 53.2 g/kg DM), followed by YQ23 (38.3 to 46.5 g/kg DM), DK301 (43.5 to 47.9 g/kg DM), ZD958 (36.0 to 44.9 g/kg DM) and YQ889 (39.8 to 44.5 g/kg DM). The rumen degradability gap between different regions could be explained by the harvest procedure, and the land irrigation or fertilization may other factors attributed to the discrepancy [45]. Cox et al. [46] reported that *in vitro* NDF digestibility of six corn hybrids on average varied from 710 to 827 g/kg between two growing regions, but limited information exists to make a comparison of *in situ* CP rumen degradation kinetics for forages cultivated in different regions. In terms of cultivars, the highest ($P < 0.05$) content of RDP was observed in DK301 (47.8 g/kg DM) and lowest in ZD958 (40.2 g/kg DM). Additionally, the highest and lowest variation of RDP content among regions was observed for FKBN (33.7 g/kg DM in Urumchi to 53.2 g/kg DM in Tianjin) and YQ889 (38.3 g/kg DM in Cangzhou to 44.5 g/kg DM in Beijing), respectively. The results indicated that the cultivar of DK301 contained higher ruminally degraded CP and this indicator was more stable in the cultivar of YQ889. The differences in the molecular structure may be the reason for the discrepancy in rumen degradable CP among the various cultivars of WPSC [47].

**Table 5. *In situ* crude protein (CP) degradation kinetics of whole plant silage corn cultivars grown in different regions.**

| Cultivars | Regions | $K_d$ (/h) | S (g/kg CP) | D (g/kg CP) | RUP(g/kg DM) | RDP (g/kg DM) |
|---|---|---|---|---|---|---|
| FKBN | Beijing | 0.080 | 479.8 | 520.2 | 40.20 | 47.30 |
| | Urumchi | 0.021 | 371.4 | 465.3 | 35.79 | 33.71 |
| | Cangzhou | 0.029 | 381.5 | 472.0 | 38.81 | 43.29 |
| | Tianjin | 0.026 | 536.9 | 206.3 | 35.89 | 53.21 |
| | Liaoyuan | 0.020 | 413.3 | 586.7 | 31.37 | 39.33 |
| YQ889 | Beijing | 0.022 | 464.4 | 281.4 | 40.26 | 44.54 |
| | Urumchi | 0.067 | 422.0 | 193.1 | 37.55 | 39.85 |
| | Cangzhou | 0.029 | 456.0 | 311.0 | 31.21 | 38.29 |
| | Tianjin | 0.006 | 443.3 | 556.7 | 40.35 | 39.75 |
| | Liaoyuan | 0.012 | 455.7 | 544.3 | 36.74 | 44.46 |
| YQ23 | Beijing | 0.015 | 431.8 | 454.6 | 44.27 | 46.53 |
| | Urumchi | 0.013 | 484.6 | 422.6 | 30.73 | 38.27 |
| | Cangzhou | 0.049 | 501.6 | 282.9 | 31.51 | 53.29 |
| | Tianjin | 0.023 | 441.9 | 346.7 | 44.68 | 46.22 |
| | Liaoyuan | 0.029 | 428.2 | 401.7 | 38.74 | 43.26 |
| DK301 | Beijing | 0.012 | 373.4 | 626.6 | 47.92 | 43.48 |
| | Urumchi | 0.013 | 463.4 | 509.7 | 36.82 | 45.78 |
| | Cangzhou | 0.063 | 409.4 | 396.4 | 34.15 | 53.75 |
| | Tianjin | 0.171 | 367.2 | 417.6 | 37.55 | 47.85 |
| | Liaoyuan | 0.016 | 489.3 | 510.7 | 32.72 | 47.88 |
| ZD958 | Beijing | 0.012 | 45.30 | 52.79 | 36.01 | 42.49 |
| | Urumchi | 0.017 | 39.45 | 60.55 | 32.80 | 36.00 |
| | Cangzhou | 0.012 | 46.73 | 53.27 | 29.45 | 37.15 |
| | Tianjin | 0.150 | 42.91 | 34.85 | 37.88 | 44.92 |
| | Liaoyuan | 0.015 | 41.49 | 58.51 | 35.11 | 40.29 |
| SEM | | 0.0718 | 0.845 | 2.422 | 0.905 | 1.040 |
| Regions | | | | | | |
| | Beijing | 0.014[b] | 44.05 | 48.21[ab] | 41.73[a] | 44.87[ab] |
| | Urumchi | 0.026[ab] | 42.72 | 43.92[ab] | 34.74[b] | 38.72[c] |
| | Cangzhou | 0.036[ab] | 44.32 | 39.90[ab] | 33.03[b] | 45.15[ab] |
| | Tianjin | 0.066[a] | 44.37 | 37.52[b] | 39.27[a] | 46.39[a] |
| | Liaoyuan | 0.019[b] | 44.03 | 52.57[a] | 34.94[b] | 43.04[b] |
| SEM | | 0.0157 | 1.158 | 4.550 | 1.127 | 1.127 |
| P-value | Region | 0.016 | 0.85 | 0.016 | <0.001 | <0.001 |
| | Cultivar | 0.59 | 0.22 | 0.13 | 0.16 | <0.001 |
| | Region × Cultivar | 0.48 | 0.002 | 0.29 | 0.06 | <0.001 |

Note: FKBN = Fengkenbainuo; YQ 889 = Yaqing 889; YQ 23 = Yuqing 23; DK 301 = Dongke 301; ZD958 = Zhengdan 958; 2 $K_d$ = ruminal degradation rate; S = soluble CP in the in situ incubation; D = insoluble but potentially degradable CP in the in situ incubation; U = potential undegradable CP in the in situ incubation; RUP = the crude protein fraction of bypass rumen; RDP = the degradable fraction of crude protein in the rumen. Means with different letters are significantly different ($P < 0.05$).

## Effects of growing regions and cultivars on protein molecular structures of whole plant silage corn

Previous studies showed that feed protein solubility, ruminal degradability and post-ruminal digestibility not only depended on CP chemical compositions, but also strongly influenced by the protein inherent molecular structures [48–50]. Therefore, differences in protein spectral characteristics of WPSC were quantified and visualized between growing regions and corn cultivars. Data on secondary protein structures showed an interaction of growing region and

**Table 6. Protein molecular structures of silage corn cultivars grown in different regions.** %.

| Cultivars | Regions | Protein Primary Structure Characteristics | | | | | | Protein Secondary Structure | | |
|---|---|---|---|---|---|---|---|---|---|---|
| | | Amide peak I height | Amide peak II height | Amide peak (I: II) height ratio | Amide peak I area | Amide peak II area | Amide peak (I: II) area ratio | $\alpha$-helix height | $\beta$-sheet height | Height ratio $\alpha$-helix: $\beta$-sheet |
| FKBN | Beijing | 0.07 | 0.01 | 5.34 | 4.79 | 0.61 | 7.87 | 0.07 | 0.06 | 1.20 |
| | Urumchi | 0.05 | 0.01 | 3.27 | 3.71 | 0.36 | 10.28 | 0.08 | 0.04 | 2.40 |
| | Cangzhou | 0.07 | 0.02 | 3.68 | 5.32 | 0.52 | 10.83 | 0.07 | 0.07 | 0.98 |
| | Tianjin | 0.06 | 0.01 | 4.30 | 5.20 | 0.39 | 13.44 | 0.07 | 0.06 | 1.16 |
| | Liaoyuan | 0.13 | 0.03 | 4.72 | 10.2 | 0.69 | 11.47 | 0.12 | 0.13 | 0.98 |
| YQ889 | Beijing | 0.05 | 0.01 | 4.45 | 3.84 | 0.33 | 11.63 | 0.05 | 0.04 | 1.38 |
| | Urumchi | 0.14 | 0.04 | 3.13 | 10.72 | 1.17 | 9.20 | 0.14 | 0.07 | 1.97 |
| | Cangzhou | 0.10 | 0.02 | 5.83 | 6.51 | 0.78 | 8.35 | 0.09 | 0.08 | 1.25 |
| | Tianjin | 0.03 | 0.005 | 6.01 | 2.41 | 0.22 | 11.1 | 0.03 | 0.03 | 1.03 |
| | Liaoyuan | 0.11 | 0.02 | 4.51 | 7.99 | 0.71 | 11.29 | 0.10 | 0.10 | 1.01 |
| YQ23 | Beijing | 0.03 | 0.002 | 20.17 | 2.58 | 0.26 | 9.97 | 0.03 | 0.03 | 0.85 |
| | Urumchi | 0.08 | 0.02 | 4.72 | 5.96 | 0.61 | 9.91 | 0.08 | 0.06 | 1.54 |
| | Cangzhou | 0.07 | 0.02 | 3.78 | 5.11 | 0.52 | 9.89 | 0.07 | 0.04 | 1.60 |
| | Tianjin | 0.11 | 0.03 | 4.09 | 8.38 | 0.82 | 10.09 | 0.10 | 0.11 | 0.92 |
| | Liaoyuan | 0.09 | 0.02 | 4.83 | 6.67 | 0.67 | 9.92 | 0.09 | 0.08 | 1.04 |
| DK301 | Beijing | 0.12 | 0.004 | 28.83 | 8.84 | 0.97 | 9.09 | 0.08 | 0.12 | 0.69 |
| | Urumchi | 0.07 | 0.01 | 4.76 | 5.55 | 0.59 | 9.21 | 0.08 | 0.07 | 1.20 |
| | Cangzhou | 0.07 | 0.02 | 4.01 | 4.78 | 0.55 | 8.67 | 0.06 | 0.06 | 1.15 |
| | Tianjin | 0.11 | 0.02 | 6.06 | 7.84 | 0.58 | 13.91 | 0.10 | 0.10 | 1.03 |
| | Liaoyuan | 0.07 | 0.02 | 3.99 | 5.2 | 0.65 | 8.04 | 0.07 | 0.06 | 1.20 |
| ZD958 | Beijing | 0.10 | 0.01 | 10.31 | 7.68 | 0.68 | 11.33 | 0.10 | 0.10 | 0.99 |
| | Urumchi | 0.09 | 0.03 | 3.51 | 7.47 | 0.77 | 9.69 | 0.09 | 0.04 | 2.29 |
| | Cangzhou | 0.13 | 0.03 | 5.05 | 9.36 | 0.84 | 12.2 | 0.12 | 0.13 | 0.97 |
| | Tianjin | 0.10 | 0.02 | 4.15 | 6.91 | 0.82 | 8.30 | 0.09 | 0.08 | 1.20 |
| | Liaoyuan | 0.10 | 0.02 | 5.65 | 7.17 | 0.47 | 15.35 | 0.09 | 0.09 | 0.98 |
| | SEM | 0.006 | 0.002 | 1.131 | 0.431 | 0.043 | 0.366 | 0.005 | 0.006 | 0.083 |
| Regions | | | | | | | | | | |
| | Beijing | 0.07[b] | 0.008[c] | 13.82[a] | 5.55[b] | 0.57 | 9.98 | 0.063[b] | 0.069[bc] | 1.02[b] |
| | Urumchi | 0.09[ab] | 0.023[a] | 3.88[b] | 6.68[ab] | 0.70 | 9.66 | 0.093[a] | 0.055[c] | 1.88[a] |
| | Cangzhou | 0.09[ab] | 0.019[ab] | 4.47[b] | 6.22[ab] | 0.64 | 9.99 | 0.083[a] | 0.074[b] | 1.19[b] |
| | Tianjin | 0.08[b] | 0.017[b] | 4.92[b] | 6.15[b] | 0.57 | 11.37 | 0.078[ab] | 0.075[b] | 1.07[b] |
| | Liaoyuan | 0.10[a] | 0.021[a] | 4.74[b] | 7.45[a] | 0.64 | 17.81 | 0.094[a] | 0.092[a] | 1.04[b] |
| | SEM | 0.006 | 0.001 | 0.857 | 0.438 | 0.050 | 2.974 | 0.006 | 0.006 | 0.117 |
| P-value | Region | 0.048 | < .001 | < .001 | 0.047 | 0.28 | 0.02 | 0.005 | < .001 | < .001 |
| | Cultivar | 0.01 | 0.01 | < .001 | 0.02 | 0.053 | 0.08 | 0.050 | 0.04 | 0.39 |
| | Region × Cultivar | < .001 | < .001 | < .001 | < .001 | < .001 | < .001 | < .001 | < .001 | 0.26 |

Note: FKBN = Fengkenbainuo; YQ 889 = Yaqing 889; YQ 23 = Yuqing 23; DK 301 = Dongke 301; ZD958 = Zhengdan 958. Means with different letters are significantly different ($P < 0.05$).

cultivar for α-helix and β-sheet height (Table 6, $P < 0.05$). However, no significant interaction was observed on the α-helix to β-sheet height ratio ($P = 0.26$). The ratio of α-helix to β-sheet height varied ($P < 0.05$) among growing regions, specifically, the highest ratio was recorded for Urumchi (1.88) and lowest for Beijing (1.02). In agreement with our findings, previous study reported that crops under water stress exhibited significantly different CP structural

features and carbohydrates spectral regions compared to crops grown in wet land [6, 8]. During the growing and development period, the plants need to absorb and accumulate nutrients by the roots from soil, and water is the key transmission medium for transportation of all nutrients. Therefore, sufficient soil water could contribute to high efficiency of nutrients transportation and promotion of cell elongation [51], which might further cause a great development of molecular makeup and conformation of biopolymers in plants.

The results of CLA and PCA analyses were presented in Fig 1. It was evident from Fig 1 that the protein structural makeup of FKBN and DK301 cultivated in Tianjin were distinguished from those grown in other regions. As for cultivars of YQ889 and YQ23, clear separate ellipses were found between Cangzhou and other growing locations. For the remaining cultivars, heavy overlaps were observed for the protein primary structural data. These results indicated that for same WPSC cultivar, internal protein molecular makeups were altered when environmental conditions changed. The effect of cultivar on inherent molecular structures was investigated on different plant materials [52], and the discrepancies were contributed by various factors [27].

## Correlation of protein molecular structural features with protein chemical profiles as well as in situ degradation parameters of whole plant silage corn

Fig 2 visualized the results of relationship between protein molecular structural features and CP chemical profiles, CNCPS subfractions as well *in situ* rumen degradation characteristics. The area ratio of amide I to II was negatively correlated with the contents of SCP ($\delta$ = -0.66; $P$ = 0.002), CP ($\delta$ = -0.61; $P$ = 0.006), NPN ($\delta$ = -0.56; $P$ = 0.004), ADICP ($\delta$ = -0.43; $P$ = 0.008) and PA$_1$ ($\delta$ = -0.38; $P$ = 0.047), and positively correlated with PB$_1$ ($\delta$ = 0.58; $P$ = 0.01). The $\alpha$-helix to $\beta$-sheet height ratio was positively correlated with PB$_2$ ($\delta$ = 0.22; $P$ = 0.048) while negatively correlated with PB$_1$ ($\delta$ = -0.46; $P$ = 0.002). In agreement with our findings, the correlation between protein molecular structures and chemical profiles was demonstrated by a previous study [53]. However, Liu et al. [54] found negative relationships between $\alpha$-helix to $\beta$-sheet height ratio with PB$_2$ ($\delta$ = -0.45; $P$ < 0.01) in wheat, corn and triticale grains, which were not in line with our current study.

The amide I to II area ratio had a negative correlation with RUP content ($\delta$ = -0.48; $P$ = 0.02). Numerous studies have reported correlations between protein molecular structural traits and ruminal degradation parameters in varieties of feedstuff [55, 56]. For instance, Zhang and Yu [55] reported relationships of amide I area ($\delta$ = 0.97; $P$ = 0.005), amide II area ($\delta$ = 0.90; $P$ = 0.036), $\alpha$-helix height ($\delta$ = 0.96; $P$ = 0.011), $\beta$-sheet height ($\delta$ = 0.98; $P$ = 0.004), in conjunction with $\alpha$-helix to $\beta$-sheet height ratio ($\delta$ = -0.99; $P$ = 0.00) and RDP in hulless barley as well bioethanol coproducts of wheat or dried distillers grains with solubles. However, Gomaa et al. [42] suggested that no relationship exist between area ratio of amide I to II and RUP in canola seeds. Although the data were very limited, these inconsistent results showed that the correlations between protein molecular structural characteristics and CP chemical profiles as well as rumen degradable characteristics were specific to some extent.

## Conclusions

In conclusion, the region-cultivar variations in morphological characteristics, protein subfractions, digestive kinetics, and molecular structure were investigated in our study. Compared with other regions, the WPSC grown in Beijing had higher CP, SCP and NPN, and the RDP was lower in Urumchi. In addition, the cultivar of DK301 represented higher and more stable CP content. These changes in protein chemical profiles and rumen digestibility were strongly associated with the alteration in protein intrinsic molecular structures (Area ration Amide I: Amide II). Further in-depth studies are required to establish the predictive model based on

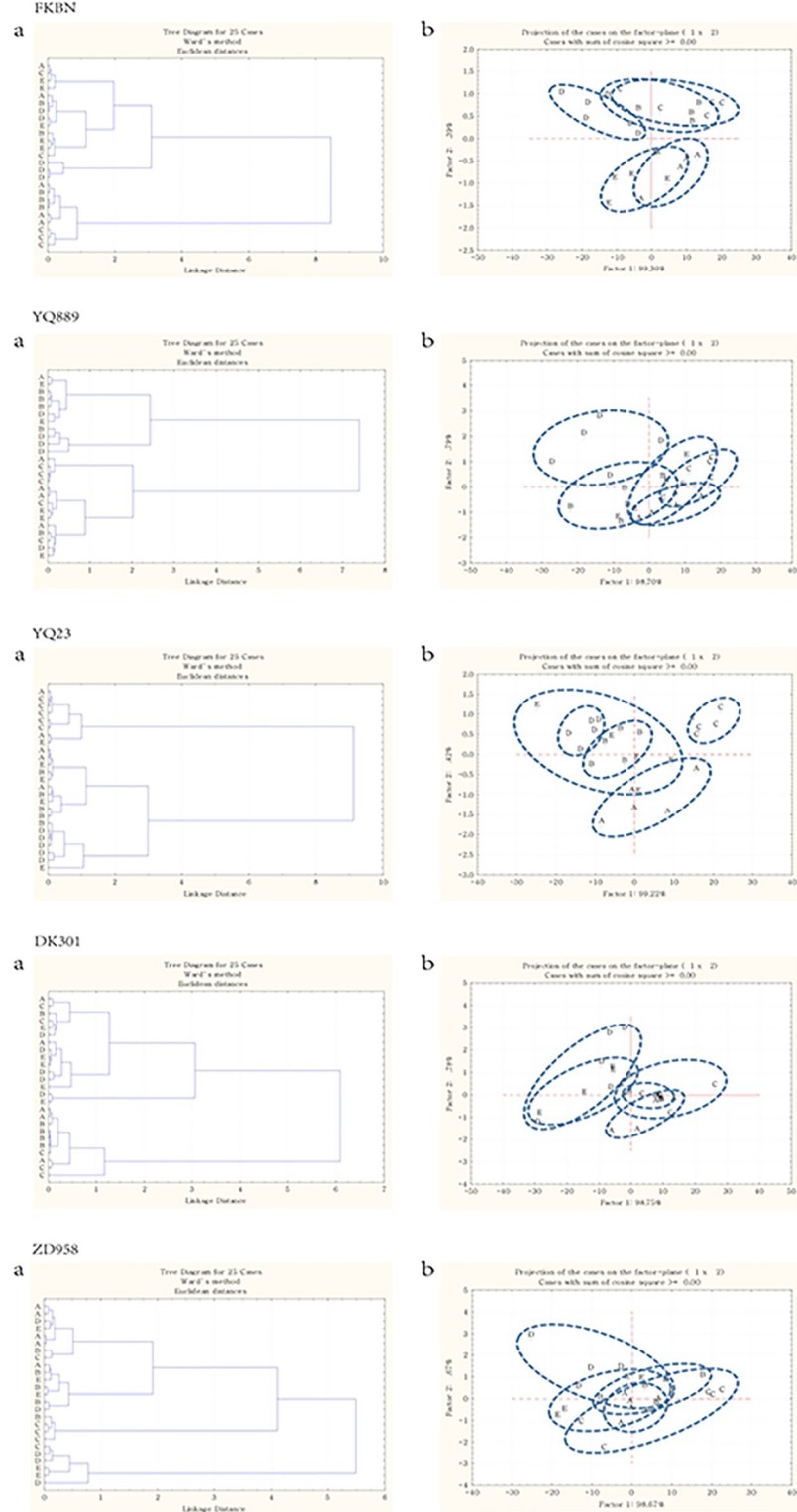

**Fig 1. Multivariate molecular spectral analyses of protein molecular spectral feature in protein amide I and amide II fingerprint region (ca. 1712–1489 cm⁻1) of various whole-plant silage corn cultivars from different regions.** a: CLA analysis; b: PCA analysis; A = Beijing; B = Urumchi; C = Cangzhou; D = Tianjin; E = Liaoyuan.

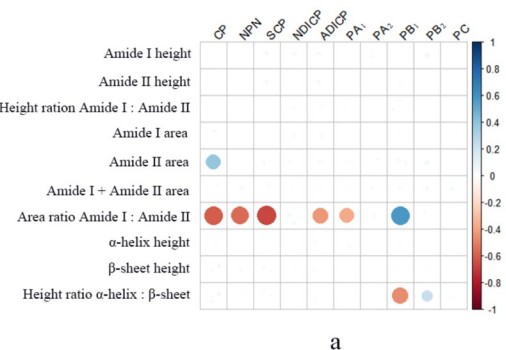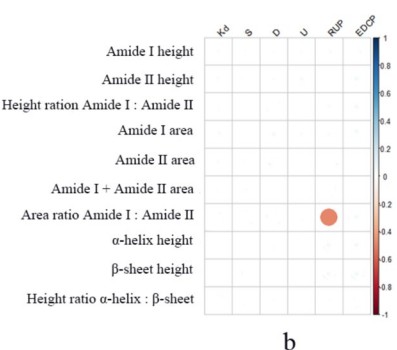

**Fig 2. Correlation of protein molecular structural features with chemical profile and CNCPS subfractions and in situ biodegradation parameters.** a, correlation of protein molecular structure features with protein chemical profile and CNCPS subfractions; b, correlation of protein molecular structure features with in situ biodegradation parameters. CP, crude protein; NPN, non-protein nitrogen; SCP, soluble CP; NDICP, neutral detergent insoluble CP; ADICP, acid detergent insoluble CP; $PA_1$, ammonia; $PA_2$, soluble non-ammonia CP; $PB_1$, moderately degradable CP; $PB_2$, slowly degradable CP; PC, unavailable CP; $K_d$, ruminal degradation rate; S, soluble CP in the in situ incubation; D, insoluble but potentially degradable CP in the in situ incubation; U, potential undegradable CP in the in situ incubation; RUP, the CP fraction of bypass rumen; EDCP, the effective degradable fraction of CP in the rumen.

novel technology (FT/IR) for efficiently estimating nutrient contents and rumen degradable characteristics from molecular spectral data.

## Acknowledgments

The authors would like to thank supplementation from Northeast Agricultural University.

## Author Contributions

**Conceptualization:** Yonggen Zhang, Hangshu Xin.

**Data curation:** Nazir Ahmad Khan, Fanlin Kong.

**Formal analysis:** Xinyue Zhang, Fanlin Kong.

**Funding acquisition:** Hangshu Xin.

**Investigation:** Enyue Yao.

**Methodology:** Xinyue Zhang, Nazir Ahmad Khan.

**Project administration:** Hangshu Xin.

**Resources:** Xinyue Zhang, Nazir Ahmad Khan.

**Software:** Enyue Yao, Ming Chen.

**Supervision:** Yonggen Zhang, Hangshu Xin.

**Validation:** Hangshu Xin.

**Visualization:** Rifat Ullah Khan, Xin Liu.

**Writing – original draft:** Xinyue Zhang.

**Writing – review & editing:** Xinyue Zhang, Nazir Ahmad Khan, Fanlin Kong.

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
