## [Decision Letter · Decision Letter 0]

30 Mar 2023

PONE-D-23-04492

Effect of growing regions on morphological characteristics, protein nutrition, rumen degradation and molecular structures of various whole-plant silage corn cultivars

PLOS ONE

Dear Dr. Zhang,

Thank you for submitting your manuscript to PLOS ONE. After careful consideration, we feel that it has merit but does not fully meet PLOS ONE’s publication criteria as it currently stands. Therefore, we invite you to submit a revised version of the manuscript that addresses the points raised during the review process.

We look forward to receiving your revised manuscript.

Kind regards,

Adham A. Al-Sagheer

Academic Editor

PLOS ONE

Journal Requirements:

2. As part of your revision, please complete and submit a copy of the Full ARRIVE 2.0 Guidelines checklist, a document that aims to improve experimental reporting and reproducibility of animal studies for purposes of post-publication data analysis and reproducibility: https://arriveguidelines.org/sites/arrive/files/documents/Author%20Checklist%20-%20Full.pdf Please include your completed checklist as a Supporting Information file. Note that if your paper is accepted for publication, this checklist will be published as part of your article.

   "This research was funded by the National Natural Science Foundation of China (Grant No. 31702135) and China Agriculture Research System of MOF and MARA."

Reviewers' comments:

Reviewer's Responses to Questions

**Comments to the Author**

1. Is the manuscript technically sound, and do the data support the conclusions?

Reviewer #1: No

Reviewer #2: Yes

Reviewer #3: Yes

Reviewer #4: Yes

Reviewer #5: No

2. Has the statistical analysis been performed appropriately and rigorously? 

Reviewer #1: I Don't Know

Reviewer #2: Yes

Reviewer #3: Yes

Reviewer #4: Yes

Reviewer #5: I Don't Know

3. Have the authors made all data underlying the findings in their manuscript fully available?

Reviewer #1: No

Reviewer #2: Yes

Reviewer #3: Yes

Reviewer #4: Yes

Reviewer #5: Yes

4. Is the manuscript presented in an intelligible fashion and written in standard English?

Reviewer #1: No

Reviewer #2: No

Reviewer #3: Yes

Reviewer #4: Yes

Reviewer #5: Yes

5. Review Comments to the Author

Reviewer #1: Lack of novelty, originality and in vivo experiments to the existing study of the field as well as regional research work in order to add to the existing knowledge unless the methodology of the work done demonstrates something new.

Reviewer #2: The manuscript evaluates the variation of morphological characteristics, nutritional value, ruminal degradability, and molecular structural makeup of diverse whole-plant silage corn cultivars among different growing regions. The study suggested that the morphological characteristics, protein nutritional values and rumen degradability significantly varied among different grown regions due to distinguished molecular structures. The manuscript is novel and important for the readers. However, some inquiries need to be clarified before the final decision.

(1) The English in the text should be improved.

(2) My main concern is the statistical analysis. Please explain in details the CLA and PCA analysis. Could you send me the code used for this analysis? what was your experimental unit for all of your obtained results? which significance degree did you use?

(3) Please insert more information about the plant collection, plant sample size? how many trees were collected for each plant sample? how many plots are for each plant sample? how many true statistical and analytical repetitions were made for all the obtained results?

(4) There may be need to know how much feed the animals were given to ensure maintenance energy intake. The same applies to the amount of treatment samples incubated for degradation, no details on the amount of samples in terms of weight per bag and the number of bags per animal.

Reviewer #3: It feels like a record of in-house experimentation at a seed company.

Isn't it natural to expect that plants with different genetic traits, grown in different climates, will show different growth processes?

I don't think this is a new scientific law.

Did you get permission from the ethics committee to conduct animal experiments? What is the permission number?

Other minor comments were redacted in the manuscript.

Reviewer #4: The topic of the manuscript is interesting and includes practical findings. However, it has issues that need to be fixed. In particular, the Materials and Methods, Discussion and Conclusion sections should be improved. Please see the annotated file for some comments.

Reviewer #5: 1.statistic difficult to understand

2.Treatment should classified more difficult to understand

3.This manuscript how many experiment difficult to understand

4.Should recheck format of journal

5.Should recheck grammar or ask English personal to proof

6. PLOS authors have the option to publish the peer review history of their article (what does this mean?). If published, this will include your full peer review and any attached files.

Reviewer #1: No

Reviewer #2: No

Reviewer #3: No

Reviewer #4: No

Reviewer #5: **Yes: **Thitima Norrapoke

---

## [Author Response · Author response to Decision Letter 0]

11 Apr 2023

Reviewer #1: 

Lack of novelty, originality and in vivo experiments to the existing study of the field as well as regional research work in order to add to the existing knowledge unless the methodology of the work done demonstrates something new.

Response: 

Thanks for the review’s comments. We apologize for the ambiguous intent in this manuscript, and the integrated logistics about this research was reorganized as follow. We sincerely hope that our logic now is clearer to exhibit novelty and originality.

On the one hand, although previous studies have demonstrated that the different nutrient components between various forages, it was not comprehensive and lack representativeness. In current study, our sample collections and detected indicators were comprehensive, which could offset the problem involved in previous studies.

On the other hand, the Chinese dairy industry has developed rapidly over recent decades and needs to provide milk products to a quarter of the world's population. This further requires sustainable supplementation of feedstuffs to adapt the intensive systems. It is important to assess the nutritional concentrations and rumen degradation of forage before it can be used in the dairy industry. However, the traditional laboratory analysis for roughage costs a mass of human, material, and financial resources, as well contributes to some errors due to different operators and instruments. Therefore, as a rapid, accurate, and portable technology, Fourier transform infrared (FTIR) spectroscopy could be applied to establish a model for predicting nutrient components and digestibility. However, to our knowledge, the whole plant silage corn samples utilized to build models was only limited in a specific site. I remember that a predictive model for nutrient components of forage established by my senior sister, who collected samples only in Beijing. So, we next plan to construct a predictive model based on the data collected across the country. Above all, this manuscript represented the first section of our work. That is the necessary to build efficient predictable model to estimate nutrients of whole plant silage corn before they were utilized because of huge variation induced by cultivars and grown regions. 

Anyway, sorry for the confusion brought the reviewer again, and we really hope this explanation will help the reviewer understand this manuscript better.

Reviewer #2: 

The manuscript evaluates the variation of morphological characteristics, nutritional value, ruminal degradability, and molecular structural makeup of diverse whole-plant silage corn cultivars among different growing regions. The study suggested that the morphological characteristics, protein nutritional values and rumen degradability significantly varied among different grown regions due to distinguished molecular structures. The manuscript is novel and important for the readers. However, some inquiries need to be clarified before the final decision.

(1) The English in the text should be improved.

(2) My main concern is the statistical analysis. Please explain in details the CLA and PCA analysis. Could you send me the code used for this analysis? what was your experimental unit for all of your obtained results? which significance degree did you use?

(3) Please insert more information about the plant collection, plant sample size? how many trees were collected for each plant sample? how many plots are for each plant sample? how many true statistical and analytical repetitions were made for all the obtained results?

(4) There may be need to know how much feed the animals were given to ensure maintenance energy intake. The same applies to the amount of treatment samples incubated for degradation, no details on the amount of samples in terms of weight per bag and the number of bags per animal.

Response: 

The statements have been corrected. We will be happy to edit the text further based on the helpful comments from the review.

(1)We apologize for the poor language of our manuscript. We worked on the manuscript for a long time and the sentences obviously led to poor readability were revised repeatedly or removed. Now, we really hope that the flow and language level have been substantially improved.

(2)Sorry for the confusion. There was no code, because the Statistica 8.0 software was used to analyze the CLA and PCA. For the cluster analyses, Ward’s Algorithm method was used for clustering calculation, with results displayed as dendrograms (Yu, 2010), and the Euclidean method was applied to the distance matrix calculation. In the Principal Component Analysis, all original variables were transferred to a set of new uncorrelated variables called principal components (PCs). Then, the first principal component (PC1) and second principal component (PC2) were generated in a scatter plot and used to describe all variables. The specific spectral analyses of these two methods were explained in Yu (2005, 2010). 

Reference:

Yu P. Applications of hierarchical cluster analysis (CLA) and principal component analysis (PCA) in feed structure and feed molecular chemistry research, using synchrotron-149 based Fourier transform infrared (FTIR) microspectroscopy. J Agric Food Chem. 2005 Sep; 53:7115-7127.

Yu P. Plant-based food and feed protein structure changes induced by gene-transformation, heating and bio-ethanol processing: a synchrotron-based molecular structure and nutrition research program. Mol Nutr Food Res. 2010 Nov; 54:1535-1545.

(3) Total five cultivars of WPCS (FKBN, YQ889, YQ23, DK301 and ZD958) were selected for this study from five different areas (Beijing, Urumchi, Cangzhou, Liaoyuan, Tianjin) of China. In each area, three plots were selected, and 20 plants from each plot, 10 cm above the ground, were randomly harvested at the kernel maturity stage of the half milk line. The exterior 1 m area of each plot was excluded from sampling to ensure uniformity in the plants being sampled. Anyway, from the point of statistic, the average value of nutrient indicators from 20 WPCS plants were seemed as raw data, three plots of per sample were treated as replicates. Sorry for the unclear description in original manuscript, we have corrected it and really hope that the new version could be more accurate.

(4) Sorry for the careless, the ingredients and nutrient contents of basal diet for dairy cows were added in this manuscript (Table 2). Each nylon bag (10 × 20 cm, pore size 40 μm) was placed 7 g WPSC, and three replicates were set for per sample at every incubation time point. The “gradual in/all out” schedule was used in this experiment, so the number of bags were changed at different time point, but the total number was no more than 30 nylon bags for each animal. Specifically, a total number of 75 nylon bags (25 samples × 3 replicates) at each time point were randomly selected for three individual dairy cows, and the nutrient degradation process was conducted twice as the random effect of statistical analysis in this trial. The detailed content was added in the material and methods section of this manuscript.

Reviewer #3: 

It feels like a record of in-house experimentation at a seed company.

Isn't it natural to expect that plants with different genetic traits, grown in different climates, will show different growth processes?

I don't think this is a new scientific law.

Did you get permission from the ethics committee to conduct animal experiments? What is the permission number?

Other minor comments were redacted in the manuscript.

Response: 

Thanks for the review’s comments. We apologize for the ambiguous intent in this manuscript, and the integrated logistics about this research was reorganized as follow. We sincerely hope that our logic now is clearer to exhibit novelty and originality.

On the one hand, although previous studies have demonstrated that the different nutrient components between various forages, it was not comprehensive and lack representativeness. In current study, our sample collections and detected indicators were comprehensive, which could offset the problem involved in previous studies.

On the other hand, the Chinese dairy industry has developed rapidly over recent decades and needs to provide milk products to a quarter of the world's population. This further requires sustainable supplementation of feedstuffs to adapt the intensive systems. It is important to assess the nutritional concentrations and rumen degradation of forage before it can be used in the dairy industry. However, the traditional laboratory analysis for roughage costs a mass of human, material, and financial resources, as well contributes to some errors due to different operators and instruments. Therefore, as a rapid, accurate, and portable technology, Fourier transform infrared (FTIR) spectroscopy could be applied to establish a model for predicting nutrient components and digestibility. However, to our knowledge, the whole plant silage corn samples utilized to build models was only limited in a specific site. I remember that a predictive model for nutrient components of forage established by my senior sister, who collected samples only in Beijing. So, we next plan to construct a predictive model based on the data collected across the country. Above all, this manuscript represented the first section of our work. That is the necessary to build efficient predictable model to estimate nutrients of whole plant silage corn before they were utilized because of huge variation induced by cultivars and grown regions. 

Anyway, sorry for the confusion brought the reviewer again, and we really hope this explanation will help the reviewer understand this manuscript better.

As for the ethics committee, our study was approved by the Ethical Committee of the College of Animal Science and Technology, Northeast Agriculture University for animal experiment (NEAU-[2011]-9).

Finally, the statements noted in this manuscript were corrected seriously step by step. We will be happy to edit the text further based on the helpful comments from the review.

Reviewer #4: 

The topic of the manuscript is interesting and includes practical findings. However, it has issues that need to be fixed. In particular, the Materials and Methods, Discussion and Conclusion sections should be improved. Please see the annotated file for some comments.

Response: 

Thanks for the review’s suggestions, and we have carefully revised the relative contents in accordance with the comments.

Reviewer #5: 

1.statistic difficult to understand

2.Treatment should classified more difficult to understand

3.This manuscript how many experiment difficult to understand

4.Should recheck format of journal

5.Should recheck grammar or ask English personal to proof

Response: 

Thanks for the review’s suggestions. According to his/her advice, we have reorganized the inappropriate contents in the new version of the manuscript. We will be happy to edit the text further based on the helpful comments from the review.

---

## [Decision Letter · Decision Letter 1]

2 May 2023

PONE-D-23-04492R1Effect of growing regions on morphological characteristics, protein subfractions, rumen degradation and molecular structures of various whole-plant silage corn cultivarsPLOS ONE

Dear Dr. Zhang,

Thank you for submitting your manuscript to PLOS ONE. After careful consideration, we feel that it has merit but does not fully meet PLOS ONE’s publication criteria as it currently stands. Therefore, we invite you to submit a revised version of the manuscript that addresses the points raised during the review process.

We look forward to receiving your revised manuscript.

Kind regards,

Adham A. Al-Sagheer

Academic Editor

PLOS ONE

**Additional Editor Comments:**

1) The manuscript contains many grammatical, typographic, and styling errors. English editing of the manuscript by a native English speaker is highly recommended.

2) The research question is not clearly outlined in the Introduction section.

3) Figure 1 has a low quality and should be presented in higher resolution.

4) The discussion is superficial. All estimated parameters should be discussed in more detail with more depth. Also, the discussion must be fortified with possible attributions.

Reviewers' comments:

Reviewer's Responses to Questions

**Comments to the Author**

1. If the authors have adequately addressed your comments raised in a previous round of review and you feel that this manuscript is now acceptable for publication, you may indicate that here to bypass the “Comments to the Author” section, enter your conflict of interest statement in the “Confidential to Editor” section, and submit your "Accept" recommendation.

Reviewer #4: (No Response)

Reviewer #5: (No Response)

2. Is the manuscript technically sound, and do the data support the conclusions?

Reviewer #4: Partly

Reviewer #5: Yes

3. Has the statistical analysis been performed appropriately and rigorously? 

Reviewer #4: (No Response)

Reviewer #5: Yes

4. Have the authors made all data underlying the findings in their manuscript fully available?

Reviewer #4: Yes

Reviewer #5: Yes

5. Is the manuscript presented in an intelligible fashion and written in standard English?

Reviewer #4: Yes

Reviewer #5: Yes

6. Review Comments to the Author

**Reviewer #4:** The manuscript has been partially improved. Some things still need to be corrected and clarified.

In the section “Corn cultivars, growing regions, and crop production”:

More clarification is needed about the practical management and agronomic practices including the soil type in each region, fertilization, irrigation, planting time (date) in each region, etc.

Soil type, temperature and rainfall are different in different regions. Was the uniform management reasonable? The case should be clarified/justified.

In the section “Crop sampling and morphological measurements”:

The harvest date in each region must be specified precisely.

In the section “In situ rumen degradation”:

At what level was the feeding done? Maintenance?

In the Discussion:

Many parts of the discussion are similar to the review of sources. This part should be improved and the results should be interpreted more carefully and the causes of differences among the regions should be justified based on the characteristics and conditions of each region.

**Reviewer #5: **The manuscript evaluates the variation of morphological characteristics, nutritional

value, ruminal degradability, and molecular structural makeup of diverse whole-plant

silage corn cultivars among different growing regions. The study suggested that the

morphological characteristics, protein nutritional values and rumen degradability

significantly varied among different grown regions due to distinguished molecular

structures. The manuscript is novel and important for the readers. 

7. PLOS authors have the option to publish the peer review history of their article (what does this mean?). If published, this will include your full peer review and any attached files.

Reviewer #4: No

Reviewer #5: No

---

## [Author Response · Author response to Decision Letter 1]

6 May 2023

Additional Editor Comments:

1) The manuscript contains many grammatical, typographic, and styling errors. English editing of the manuscript by a native English speaker is highly recommended.

2) The research question is not clearly outlined in the Introduction section.

3) Figure 1 has a low quality and should be presented in higher resolution.

4) The discussion is superficial. All estimated parameters should be discussed in more detail with more depth. Also, the discussion must be fortified with possible attributions.

Response: 

Thanks for your suggestions and we have carefully revised the relative contents in accordance with the comments.

Firstly, we rechecked the manuscript by a native English speaker at your suggestion. 

Then, the introduction and discussion sections were revised appropriately. For introduction part, we give a brief introduction of whole-plant silage corn (WPSC) and its important role in first paragraph, and the environmental factors for WPSC growth were discussed in second paragraph. Then, the association of protein chemical profiles and in situ degradation parameters with the spectral data of specific inherent structures were analyzed. In terms of discussion part, some new references were inserted in the manuscript to discuss more depth. 

Finally, a clearer Figure 1 was reuploaded. 

Anyway, thank you again, and we all be happy to edit the text further based on the helpful comments from you!

Reviewer #4: 

The manuscript has been partially improved. Some things still need to be corrected and clarified.

In the section “Corn cultivars, growing regions, and crop production”:

More clarification is needed about the practical management and agronomic practices including the soil type in each region, fertilization, irrigation, planting time (date) in each region, etc. Soil type, temperature and rainfall are different in different regions. Was the uniform management reasonable? The case should be clarified/justified.

In the section “Crop sampling and morphological measurements”:

The harvest date in each region must be specified precisely.

In the section “In situ rumen degradation”:

At what level was the feeding done? Maintenance?

In the Discussion:

Many parts of the discussion are similar to the review of sources. This part should be improved and the results should be interpreted more carefully and the causes of differences among the regions should be justified based on the characteristics and conditions of each region.

Response: 

Thanks for your suggestions and we have carefully revised the relative contents in accordance with the comments.

1)In the section “Corn cultivars, growing regions, and crop production”:

The planting information as well regional temperature and precipitation were represented in text and Table 1, respectively. The planting time (date) was not mentioned because the samples were harvested at the same stage (kernel maturity stage of half milk line) whenever they were planted. The uniform management is unreasonable, so we suppled the sentence “The fertilization and irrigation were managed by the local farmers accordingly” in manuscript for better explanation. As for the relative information of soil, we are really SORRY for that. Because of some external factors, the collected soil samples were broken and could not be used anymore. We tried our best to measure, but the data was not ideal. Finally, we gave up the soil information to ensure the accuracy of data. We are eager for your understanding. 

2)In the section “Crop sampling and morphological measurements”:

We must explain honestly for that. The specific harvest date for each region was ambiguous. But we promise that all samples were harvested from late September to early October at kernel maturity stage of half milk line.

3)In the section “In situ rumen degradation”:

The ingredients and nutrient contents of TMR are list in Table 2 of manuscript according to recommendations in NRC (2001) guidelines. We have added the reference in the new version, sorry for the careless.

Reference:

NRC. Nutrient requirements of dairy cattle. 7th rev ed. 2001. Natl Acad Sci, Washington, DC.

4)In the Discussion:

We totally agree with you, so some new references were inserted in the discussion section of this latest manuscript to discuss more depth.

Anyway, thank you again, and we all be happy to edit the text further based on the helpful comments from you!

Reviewer #5: The manuscript evaluates the variation of morphological characteristics, nutritional value, ruminal degradability, and molecular structural makeup of diverse whole-plant silage corn cultivars among different growing regions. The study suggested that the morphological characteristics, protein nutritional values and rumen degradability significantly varied among different grown regions due to distinguished molecular structures. The manuscript is novel and important for the readers. 

Response: 

Thanks for your affirmation, we would like to publish this article in “PLoS One” so much.

---

## [Decision Letter · Decision Letter 2]

2 Jun 2023

Effect of growing regions on morphological characteristics, protein subfractions, rumen degradation and molecular structures of various whole-plant silage corn cultivars

PONE-D-23-04492R2

Dear Dr. Zhang,

We’re pleased to inform you that your manuscript has been judged scientifically suitable for publication and will be formally accepted for publication once it meets all outstanding technical requirements.

Kind regards,

Adham A. Al-Sagheer

Academic Editor

PLOS ONE

Additional Editor Comments (optional):

Reviewers' comments:

Reviewer's Responses to Questions

**Comments to the Author**

1. If the authors have adequately addressed your comments raised in a previous round of review and you feel that this manuscript is now acceptable for publication, you may indicate that here to bypass the “Comments to the Author” section, enter your conflict of interest statement in the “Confidential to Editor” section, and submit your "Accept" recommendation.

Reviewer #4: All comments have been addressed

2. Is the manuscript technically sound, and do the data support the conclusions?

Reviewer #4: Yes

3. Has the statistical analysis been performed appropriately and rigorously? 

Reviewer #4: (No Response)

4. Have the authors made all data underlying the findings in their manuscript fully available?

Reviewer #4: Yes

5. Is the manuscript presented in an intelligible fashion and written in standard English?

Reviewer #4: Yes

6. Review Comments to the Author

Reviewer #4: (No Response)

7. PLOS authors have the option to publish the peer review history of their article (what does this mean?). If published, this will include your full peer review and any attached files.

Reviewer #4: No

---

## [Editor Report · Acceptance letter]

29 Dec 2023

PONE-D-23-04492R2 

PLOS ONE

Dear Dr. Zhang, 

I'm pleased to inform you that your manuscript has been deemed suitable for publication in PLOS ONE. Congratulations! Your manuscript is now being handed over to our production team.

Kind regards, 

on behalf of

Dr. Adham A. Al-Sagheer 

Academic Editor

PLOS ONE